# A Comparative Study of Binding Interactions between Proteins and Flavonoids in *Angelica Keiskei*: Stability, *α*-Glucosidase Inhibition and Interaction Mechanisms

**DOI:** 10.3390/ijms24076582

**Published:** 2023-04-01

**Authors:** Rui Wang, Lanlan Tu, Daodong Pan, Xinchang Gao, Lihui Du, Zhendong Cai, Jinhong Wu, Yali Dang

**Affiliations:** 1State Key Laboratory for Managing Biotic and Chemical Threats to the Quality and Safety of Agro-products, College of Food and Pharmaceutical Sciences, Ningbo University, Ningbo 315211, China; 2Department of Food Science and Engineering, School of Agriculture and Biology, Shanghai Jiao Tong, Shanghai 200240, China; 3Department of Chemistry, Tsinghua University, Beijing 100084, China

**Keywords:** stability, flavonoids, protein-based nanocomplex, interaction mechanism, *A. keiskei*

## Abstract

Flavonoids are easily destroyed and their activity lost during gastrointestinal digestion. Protein-based nanocomplexes, a delivery system that promotes nutrient stability and bioactivity, have received increasing attention in recent years. This study investigated the stability, inhibitory activity against α-glucosidase and interaction mechanisms of protein-based nanocomplexes combining whey protein isolate (WPI), soybean protein isolate (SPI) and bovine serum albumin (BSA) with flavonoids (F) from *A. keiskei* using spectrophotometry, fluorescence spectra and molecular docking approaches. The results show that the flavonoid content of WPI-F (23.17 ± 0.86 mg/g) was higher than those of SPI-F (19.41 ± 0.56 mg/g) and BSA-F (20.15 ± 0.62 mg/g) after simulated digestion in vitro. Furthermore, the inhibition rate of WPI-F (23.63 ± 0.02%) against α-glucosidase was also better than those of SPI-F (18.56 ± 0.02%) and BSA-F (21.62 ± 0.02%). The inhibition rate of WPI-F increased to nearly double that of F alone (12.43 ± 0.02%) (*p* < 0.05). Molecular docking results indicated that the protein-flavonoids (P-F) binding occurs primarily through hydrophobic forces, hydrogen bonds and ionic bonds. Thermodynamic analysis (ΔH > 0, ΔS > 0) indicated that the P-F interactions are predominantly hydrophobic forces. In addition, the absolute value of ΔG for WPI-F is greater (−30.22 ± 2.69 kJ mol^−1^), indicating that WPI-F releases more heat energy when synthesized and is more conducive to combination. This paper serves as a valuable reference for the stability and bioactivity of flavonoids from *A. keiskei*.

## 1. Introduction

*Angelica keiskei* (*A. keiskei*) is a medicinal and edible plant with a wide range of pharmacological effects. Studies have shown that *A. keiskei* contains various aurones, flavanols, coumarins, and chalcones [1]. Its stems and leaves are often consumed as a healthy vegetable because of their anti-cancer, blood pressure-lowering, and blood sugar-regulating effects, among others [2]. However, the poor stability and bioavailability of flavonoids (F) from *A. keiskei* hinder their application in functional foods [3]. It has been reported that less than 17% and 1% of single flavonoids can be orally absorbed by animals and humans, respectively [4]. Single flavonoids have low solubility in water (<8 μg/mL). In addition, the oral bioavailability of single flavonoids is not high, as it has a concentration of less than 6 μg/mL in simulated gastric digestive fluid and of less than 29 μg/mL in simulated intestinal digestive fluid [5]. Accordingly, protein-based nanocomplexes was developed to enhance the stability and bioavailability of these flavonoids. Protein-based nanocomplexes have become a research hot spot in recent years due to a number of promising characteristics, including high solubility, emulsification potential, stability and bioavailability. A variety of small molecule drugs and nutrients have been bound to natural proteins as mixed systems for study [4,6,7]. For example, the encapsulation of curcumin in α-lactalbumin nanocomplexes can improve its functional activity [8].

Whey protein isolate (WPI) is widely used as a natural protein for nutrient encapsulation and delivery, because it is easily absorbed and provides nutrients, anti-pepsin activity, and other advantages for the human body [9]. Bovine serum albumin (BSA), a carrier with high structural homology to human serum albumin (76% resemblance), has the advantages of high stability and good physical characteristics [10]. A BSA-dextran conjugate has been used as an emulsifier and stabilizer to protect the loaded curcumin from decomposition [11]. Soybean protein isolate (SPI) has good emulsification, solubility and other functional properties, leading to its use by many researchers as a delivery and encapsulation material for various polyphenolic compounds [12]. SPI, WPI, and BSA can all be used as natural protein materials to increase the bioavailability of nutrients. However, the use of protein-based nanocomplexes as a delivery system that promotes the stability and bioactivity of flavonoids from *A. keiskei* has yet to be comprehensively analyzed, sparking the research described in this paper.

To date, multispectral, thermodynamic, and molecular docking analyses, among other methods, have been used to analyze the complex interactions between proteins and flavonoids [13], including molecular dynamics simulations (MDS) and Fourier transform infrared spectroscopy (FTIR). It has been reported that hydrophobic interactions, hydrogen bonding and van der Waals interactions play important roles in the binding of proteins and flavonoids [14,15,16]. Thermodynamic and fluorescence spectroscopic analyses, respectively, showed that hydrophobic forces and hydrogen bonding are the primary driving forces of WPI-F and that tryptophan residues in WPI contribute to hydrogen bonding [17]. FTIR spectroscopy suggests that black bean protein and quercetin complexes include many α-helix and β-sheet structures, and complexes demonstrate better emulsifying properties than black bean protein alone [18]. The binding of proteins and polyphenols may make β-Lg structures less flexible [19]. 

The present study aimed to improve the stability and biological activity of flavonoids through their bonding with SPI, WPI, and BSA. In addition, the related mechanism of protein-flavonoids (P-F) was investigated through fluorescence spectra and molecular docking, with the ultimate goal of providing a theoretical basis for the functional application of flavonoids from *A. keiskei*.

## 2. Results and Discussion

### 2.1. Analysis of the Flavonoid Composition of A. keiskei Extract

The total flavonoid content was determined using the colorimetric method. A sample of 30 mg powder was dissolved in a 25 mL volumetric bottle, and the contents before and after purification were 12.15 ± 0.12% and 36.61 ± 0.15%, respectively. After purification by HPD-600 macroporous resin, the flavonoid content increased by 3.01 folds.

The constituents and relative contents of flavonoids from the ethanol extract of *Angelica keiskei* were analyzed by ultraperformance liquid chromatography-tandem mass spectrometry (UPLC-Q/TOF-MS). The main active constituents are chalcones, coumarins, and flavanones [20]. As shown in Table 1, flavonoid compounds with high content in the last column included aureusidin-4-O-glucoside (1), kaempferol-3-O-glucoside (2), luteolin-7-O-rutinoside (3), kaempferol-3-O-neohesperidoside (4), luteolin-7-O-neohesperidoside (5), and kaempferol-3-O-glucorhamnoside (6). Chemical structures are shown in Figure 1. These six flavonoids were selected as representative subjects for molecular docking with SPI, WPI, and BSA to study the interactions between proteins and flavonoids.

### 2.2. The Influence of the Combination of Flavonoids and Proteins on Flavonoid Bioavailability at Different Digestive Stages

Recent studies have revealed that flavonoid bioavailability is significantly increased in the presence of proteins [11,21]. Therefore, it was very important for us to investigate the binding interactions of *A. keiskei* flavonoids with proteins before further studying its biological functions. Figure 2A shows the changes in flavonoid content for samples combined with different proteins. The samples were combined with different proteins (SPI, WPI, and BSA) by nanobonding. The results showed that the F contents of the SPI, WPI, and BSA groups were high, reaching levels of 311.33 ± 0.67 mg/g, 352.38 ± 3.34 mg/g, and 353.33 ± 2.29 mg/g, respectively (*p* < 0.05), showing almost no difference from the group without protein. Compared to the group without protein, flavonoid levels were reduced in the SPI group. These results indicated that WPI and BSA can combine well with flavonoids.

As shown in Figure 2B, the flavonoid content decreased sharply during gastric and intestinal digestion. After oral digestion, about 39.77 ± 0.13% of flavonoids were consumed from the solution containing unencapsulated sample. However, flavonoid content was further reduced by 82.26 ± 0.09% after stomach digestion. The intestinal digestion phase further depletes flavonoid content, leaving only 4.54 ± 0.17%. Therefore, flavonoid compounds need to be protected during digestive stages to enhance their stability and bioavailability.

Figure 2C shows the flavonoid content under gastric digestion. Our results showed that more than 90% of unencapsulated sample was expended by the end of gastric digestion. Nevertheless, the groups in which flavonoids had been bonded to protein nanocomplexes, specifically the WPI and BSA groups, retained higher flavonoid levels after exposure to gastric juice for 2 h. Figure 2D displays the flavonoid content of different solutions over 2 h of intestinal tract digestion. The results show that a significant proportion of flavonoids were destroyed, and the flavonoid content obviously decreased after digestion. Compared to unencapsulated flavonoids, 56.21 ± 0.01%, 86.48 ± 0.08%, and 62.16 ± 0.05% of F was retained from the SPI-F, WPI-F, and BSA-F nanoencapsulation solutions, respectively, after exposure to intestinal juice for 2 h. This protective effect may be related to the combination of flavonoids and proteins [21]. After gastrointestinal tract digestion, the content of flavonoid compounds in the protein group was higher than that of the no protein group. These experiments demonstrate that different protein delivery systems can be used effectively to load and protect flavonoids. Hence, encapsulated flavonoids have greater stability than free flavonoids.

### 2.3. Inhibitory Effect against α-Glucosidase

As shown in Figure 3, the inhibitory effects of flavonoids against α-glucosidase decreased substantially for the no protein group after overall digestion. After flavonoids are digested, the inhibition of α-glucosidase is weakened. This is due to the fact that the flavonoid plane structure is changed during digestion, destroying conjugate bonds and making it difficult for flavonoid compounds to enter the enzyme capsule [22]. Herein, we investigated the α-glucosidase inhibitory effects of flavonoids in complex with three proteins, and the results are presented in Figure 3. For α-glucosidase, the inhibitory activity of flavonoids was remarkedly strengthened by the addition of SPI, WPI, or BSA. Nanoencapsulation technology can enhance the water solubility, thermal stability, gastrointestinal stability and bioavailability of biologically active compounds [23]. Specifically, WPI-F demonstrated the most dramatic inhibitory effect against α-glucosidase, reaching 23.63 ± 0.02% (*p* < 0.05) over simulating digestion. Overall, this decreased inhibition is most likely associated with the decreased flavonoid content, as shown in Figure 3 [24].

### 2.4. Molecular Docking Analysis

Molecular docking methods are widely used to study the binding sites and interaction forces of proteins and other small molecule compounds. We analyzed the forces associated with the docking of SPI, WPI, and BSA to our six candidate flavonoids in order to better understand the interactions between proteins and typical flavonoid compounds. The docking results not only reveal the three-dimensional and two-dimensional structures of proteins and flavonoids but also provide information about hydrogen bonding and hydrophobic interactions. As shown in Figure 4, hydrophobic interactions and hydrogen bonds were predominant between proteins and flavonoids.

7S and 11S are two main components of SPI. Figure 4A,B shows the docking structures of flavonoids with 7S and 11S. Beyond these interactions, the ligand appears to interact with SPI through a large number of hydrophobic interactions. Pro101, Ala164, Glu229, and Arg356 of 7S and Arg147, Asp170, Lys192, and Cys220 of 11S were found to be favorable to hydrophobic interactions, as shown in Figure 4A,B. As shown in Figure 4C, the docking results revealed that kaempferol-3-O-glucorhamnoside interacts with Ala1103, Pro1107, Trp1096 and Pro1128 of BSA through hydrophobic interactions, as well as hydrogen bonds with Arg929, Arg932, and Lys1070. Therefore, we conclude that the hydroxyl groups of flavonoid compounds can form powerful interactions with BSA. Figure 4D,E reveals that a number of amino acid residues (Leu39, Val41, Lys60, Lys69, Asn90, Met107, and Glu114 of β-Lg and Glu49, Lys58, and Tyr103 of α-La) contribute to the formation of hydrophobic forces during this interaction. The interaction between luteolin-7-O-neohesperidoside and β-Lg generates multiple hydrophobic forces, more so than other proteins. Specifically, residues Asn88 and Ser116 of β-Lg and His32, Thr33, Asn44, Asn56, and Ala106 of α-La form stable hydrogen bonds with luteolin-7-O-neohesperidoside and luteolin-7-O-rutinoside. Other interactions are described in Appendix A.

The value of “docking energy" and "docking interaction energy” can reflect the energy required in the process of interaction to a certain extent. The smaller the value is, the easier the docking is. As shown in Table 2, the average required energy for interaction between flavonoid monomer and 3VO3, 1HFZ, 3NPO, 1O5D and 3AUP is −8.1215, −17.9112, −18.2623, −20.8884 and −9.5047 kJ mol^−1^, respectively. According to 2.7, on the whole, flavonoid monomers require lower energy to interact with WPI and SPI, indicating easier binding.

### 2.5. Fluorescence Quenching Analysis 

Figure 5A1–C1 show the results of SPI, WPI and BSA fluorescence spectra, respectively, upon quenching by flavonoids. We found that these interactions can lead to fluorescence quenching and changes to concentration gradient patterns. The intrinsic fluorescence spectrum of the pure protein exhibited a shoulder peak in the range of 300–410 nm and a maximum of 329 nm. The maximum fluorescence intensity of the protein decreased with increasing flavonoid concentrations. This phenomenon indicates that flavonoids are most likely quenching the intrinsic protein fluorescence by occupying hydrophobic amino acid residues [25], indirectly suggesting the existence of a certain amount of hydrophobic interactions between proteins and flavonoids.

The functional relationship between F0/F and F is shown in Figure 5A2–C2, and the quenching constants at 18, 23, and 28 °C were respectively analyzed and obtained. Table 3 shows that the formation of SPI-F, WPI-F and BSA-F complexes was demonstrated. The maximum emission wavelength is essentially kept within a certain range of fluorescence quenching. As shown in Table 3, Kq values seem to significantly exceed the limiting diffusion rate constant of the biomolecules (2 × 10^10^ L mol^−1^ s^−1^), which illustrates that the fluorescence quenching mainly arises from static quenching by complex formation [26,27]. 

The binding of flavonoids to proteins is the result of multiple interaction forces, including electrostatic interactions, van der Waals forces, hydrophobic interactions, and hydrogen bonding [28]. As can be seen from Table 3, ΔG < 0, ΔH > 0, and ΔS > 0. These values indicate that the binding process is spontaneous, which also means that the main binding force between these flavonoids and proteins is hydrophobic interactions [29]. On the other side, the mean absolute value of ΔG for WPI-F (−30.22 ± 2.69 kJ mol^−1^) is greater than those obtained for SPI-F (−24.94 ± 1.15 kJ mol^−1^) and BSA-F (−29.66 ± 1.43 kJ mol^−1^), indicating that WPI-F releases more heat energy when synthesized and is more conducive to being combined, consistent with Table 3. In addition, it can be concluded that the binding constant has a linear relationship with temperature, such that the higher the temperature, the higher the binding constant. These results indicate that, over a certain range, higher temperatures are beneficial to enhance the mutual binding of P-F. 

## 3. Materials and Methods

### 3.1. Materials

*A. keiskei* was provided by Sichuan Moru Biotechnology Co., Ltd. (Chengdu, China) WPI, SPI, and BSA were purchased from Sigma Chemical (St. Louis, MO, USA). Acarbose was purchased from Yuanye Biotechnology Co., Ltd. (Shanghai, China). A-Glucosidase (43.6 U/mg and 4-nitrophenyl-α-d-glucopyranoside (pNPG) were obtained from Beijing Solarbio Science & Technology Co., Ltd. (Beijing, China). Bile bovine, pepsin (≥2000 U/mL), pancreatin (100 U/mL), and lipase (L3126, from porcine pancreas type II, 100–500 units/mg protein) were obtained from Sigma Chemical Company. HPD-600 macroporous adsorption resin was purchased from Nanjing Jiancheng Bioengineering Institute. (Nanjing, China). All other chemicals were of analytical grade.

### 3.2. Sample Preparation

#### 3.2.1. Extraction of Flavonoids

Samples of *A. keiskei* were cleaned, subjected to vacuum freeze-drying for 48 h, and crushed to obtain dry powder. The sample powder was mixed with 50% ethanol at a mass-to-volume ratio of 1:10 (g:mL), then heated in a water bath at 37 °C for 3 h. After the solution was centrifuged (4 °C, 8000 rpm/min), ethanol extraction was performed in triplicate. Column chromatography was performed using HPD-600 macroporous resin. The eluted ethanol-water (80:20, *v*/*v*) solution was collected and freeze-dried. Samples were stored at −20 °C until analysis.

#### 3.2.2. Flavonoid Content Assays

The flavonoid content of *A. keiskei* was determined according to the AlNO_3_ colorimetric method. First, 20 μL 3.0% NaNO_2_ solution was added to 40 μL of pre-diluted sample and mixed thoroughly and evenly. After 6 min, 20 μL 6.0% AlNO_3_, 140 μL 4.0% NaOH and 60 μL 70.0% ethanol were added and mixed in succession. After 15 min, the absorbance value of the reaction solution was measured at 510 nm. Standard curves were constructed using a rutin standard solution. Flavonoid content was expressed as the rutin equivalent per gram of dry sample weight [30].

### 3.3. Characterization of A. keiskei Flavonoids by UPLC-Q/TOF-MS

UPLC analysis conditions were as follows. The column was an Agilent SB-C18 (1.8 µm, 2.1 mm × 100 mm). The mobile phase was water (buffer A) and methanol containing 0.1% formic acid (buffer B). Gradient conditions were maintained at 5% B, 95% A for 1 min before initiation of a linear gradient to 5% A, 95% B over 9 min. The last 5% A and 95% B fractions were maintained for 1 min. Analysis was performed at a flow rate of 0.3 mL/min. The temperature of the column oven was 40 °C. The injection volume was 3 μL.

LIT and QQQ scans were performed using a three- or four-pole linear ion trap mass spectrometer (Q Trap) with ionization in electrospray ionization (ESI) mode controlled by Analyst 1.6.1 software (AB Sciex). ESI source operating parameters included an ion source temperature of 550 °C. Positive mode (5500 V) and negative mode (−4500 V) were selected from the ionization of each flavonoid by peak area. Ion source gas I (GSI) was set to 50 psi, gas II (GSII) was set to 60 psi, and curtain gas (CUR) was set to 25.0 psi. The instrument was adjusted with 10 μmol/L and 100 μmol/L polypropylene glycol solution in QQQ and LIT modes, respectively. 

### 3.4. Preparation of Complexes with A. keiskei Flavonoid Extract

To prepare the flavonoid-protein complexes, original protein emulsions were prepared by dissolving 50 mg SPI, WPI, or BSA in 100 mL water. Flavonoid stock solution was prepared by dissolving 50 mg of sample in 10 mL 50% absolute ethanol and shaking to combine. Next, 1 mL of flavonoid stock solution was slowly injected into the protein emulsion with constant stirring. Flavonoid complexes were obtained after magnetic stirring of the mixed solution in a water bath at 18, 23, or 28 °C for 60 min. Volumetric bottles containing F delivery solution were covered with aluminum foil to avoid degradation due to light exposure [31].

### 3.5. In Vitro Gastrointestinal Digestion

Simulated in vitro gastrointestinal digestion model was performed using a modified version of the following protocol [32]. To simulate gastric digestion, flavonoids or flavonoid complexes were prepared and preheated in an incubator (37 °C) for 3 min. Simulated gastric fluid (SGF) and simulated intestinal fluid (SIF) were composed as previously described. Pre-warmed flavonoid solution (10 mL) was slowly added to simulated gastric juice (SGF) buffer at a 1:1 ratio under continuous stirring. After the pH was adjusted to 2.0, simulated gastric digestion was performed. For simulated intestinal digestion, the resulting stomach sample mixture was added to an equal volume of simulated intestinal fluid (SIF) buffer and mixed thoroughly. Finally, the pH of the intestinal sample mixture was returned to 7.0. The entire simulated gastrointestinal digestion process was performed at 37 °C and 200 rpm for 2 h. After the completion of in vitro simulated gastrointestinal digestion, the reaction was terminated by incubation at −18 °C for 2 min, and the reaction products were used for subsequent experiments [33].

### 3.6. In Vitro α-glucosidase Inhibitory Assay

On the basis of a previously reported method [34], the α-glucosidase inhibitory activity of flavonoids was analyzed. Sample solution (8 μL) and α-glucosidase solution (20 μL, 1 U/mL) were added to 100 μL of 10 mmol/L phosphate buffer (pH 6.8). This mixture was thoroughly mixed and incubated at 37 °C for 15 min. After incubation, 20 μL of 2.5 mmol/L pNPG solution was added, and the mixture was incubated again at 37 °C for 15 min. After the second incubation, 80 μL sodium carbonate solution (0.2 mol/L) was added and mixed. The concentrations of F (*A. keiskei* flavonoid extract) were 50, 100, 200, 300, 400 and 500 μg/L, with three groups set up for each gradient and the average values calculated. The absorbance was recorded at 405 nm. The inhibition rate of α-glucosidase was calculated according to the following formula.
α-glucosidase inhibitory rates (%) = [1 − (A1 − A2)/(A3 − A4)] × 100

A1 and A3 represent the absorbances of the sample and of distilled water, respectively, containing α-glucosidase. A2 and A4 are defined as the absorbances of the sample and of water, respectively, in the absence of α-glucosidase. Acarbose was used as a positive control.

### 3.7. Molecular Docking

The crystal structures of α-La (PDB ID:1HFZ), β-Lg (PDB ID: 3NPO), BSA (PDB ID: 3VO3), soybean 7S protein (PDB ID: 3AUP) and 11S protein (PDB ID: 1OD5) were downloaded from the Protein Data Bank (https://www.rcsb.org/ accessed on 16 August 2022) [35]. Flavonoid structures were drawn using ChemDraw 16.0. AutoDock v4.2 (Scripps Research Institute) was used to study molecular docking. Possible conformations of ligand molecules and macromolecules were determined based on the Lamarckian genetic algorithm (LGA). In order to reduce the occurrence of randomness, optimal P-F ranking orders were selected to test the docking in triplicate. Finally, molecular docking outputs were rendered using Discovery Studio 4.5 (Accelrys Inc., San Diego, CA, USA) to visualize the conformations of 2D and 3D interaction modes [36].

### 3.8. Characterization of Complexes Containing A. keiskei Flavonoid Extract and Different Proteins

To study the interaction mechanisms of P-F (*A. keiskei* flavonoids in complex with different proteins), SPI, WPI, and BSA were selected to represent different proteins commonly found in plants and animals. Protein solutions (0.5 mg/mL) were prepared in 10 mM phosphate buffer (pH 7.0), and flavonoid extract was added to concentrations of 0, 0.01, 0.02, 0.03, 0.04, and 0.05 mg/mL. The mixed complex solutions were prepared and kept at 18, 23, or 28 °C for 30 min. Fluorescence spectra were recorded on an Agilent Cary Eclipse fluorescence spectrophotometer (Santa Clara, CA, USA), with some modifications. An excitation wavelength of 280 nm, an emission wavelength range of 310−450 nm, and a slit width of 2.0 nm (for both excitation and emission) were used. The static quenching mechanism in the linear range of the Stern-Volmer curve, the binding constant (Ka) of the complexes formed and binding site numbers (n) were calculated using the double logarithmic Stern-Volmer Equations (1) and (2) [37]. Thermodynamic parameters were calculated by the Van’t Hoff formulation (Equation (3)) and Gibbs-Helmholtz equation (Equation (4)) [38].
F_0_/F = 1 + kqτ_0_ [F] = 1 + k_SV_ [F](1)
Log [(F_0_ − F)/F] = logKa + nlog [F](2)
where F_0_ and F are the corrected protein intensities of unloaded and loaded flavonoids, respectively. [F] is the flavonoid concentration. τ_0_ is the average life span of lone protein molecules (10^−8^ s) [26]. Ksv is the Stern-Volmer quenching constant. Kq is the bimolecular quenching rate constant.
InKa = −ΔH/(RT) + ΔS/R(3)
ΔG = ΔH − TΔS(4)
where, ΔH, ΔG, and ΔS represent changes in enthalpy, Gibbs free energy, and entropy, respectively. R is the universal gas constant (8.314 J mol^−1^ K^−1^), and T is the temperature at the time of the experiment (18, 23, or 28 °C). ΔH was obtained by plotting lnKa vs. 1/T and can be regarded as constant for small temperature differences. 

### 3.9. Statistical Analysis

Experimental data results are expressed as mean ± standard deviation (SD). The software SPSS Statistics 23.0 was used to perform one-way ANOVA and Duncan variance and significance analyses. *p* < 0.05 was considered statistically significant. Origin 2020 software was used for mapping.

## 4. Conclusions

In this paper, the stability and interaction mechanisms of SPI-F, BSA-F, and WPI-F were studied based on total flavonoid levels before and after digestion, fluorescence spectra and molecular docking methods. Analysis of flavonoid content before and after simulated digestion showed that SPI-, WPI-, and BSA-based nanocomplexes all have protective effects on flavonoid compounds, with WPI demonstrating the best effect. In addition, SPI-, WPI-, and BSA-based nanocomplexes can also enhance the bioactivity of flavonoid compounds during simulated digestion. The inhibitory rate against α-glucosidase was highest for the WPI group, indicating that the inhibition by flavonoids of α-glucosidase may be related to its levels. In terms of protein and flavonoid interactions, fluorescence quenching results showed that the interaction was a spontaneous process driven mainly by hydrophobic forces. WPI-F releases the most free energy during the process of interaction, indicating that WPI and F are most easily combined. From the molecular docking results, we determined that hydrophobic interactions and hydrogen bonds are the main interaction forces underlying the binding of flavonoids to corresponding protein sites. We identified more hydrophobic forces and hydrogen bonds for WPI-F, which is in accordance with the results of fluorescence spectrum assay analysis. These results suggest that WPI-based nanocomplexes can better encapsulate and deliver flavonoids from *A. keiskei*. This study provides detailed information on the optimal flavonoid encapsulation efficiency of proteins in flavonoid-protein nanocomplexes. The flavonoid mixture used in this study is more suitable for practical production applications. Further future studies should be conducted to demonstrate the interactions of single flavonoids with *A. keiskei* proteins and to promote their bioavailability.

## Figures and Tables

**Figure 1 ijms-24-06582-f001:**
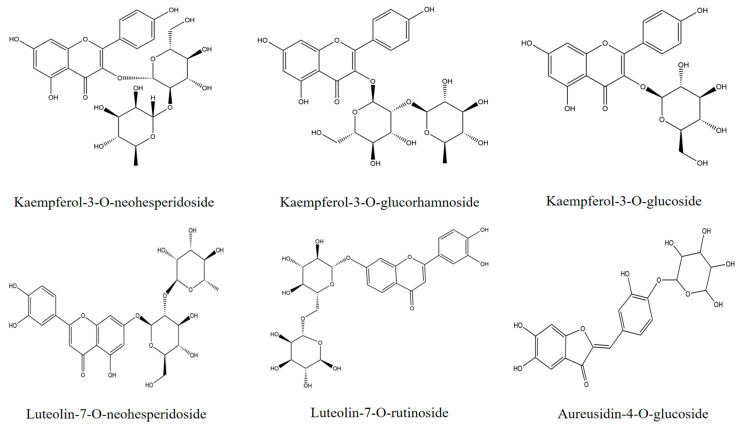
Chemistry structures of the six flavonoids in terms of content in the *A. keiskei* extraction.

**Figure 2 ijms-24-06582-f002:**
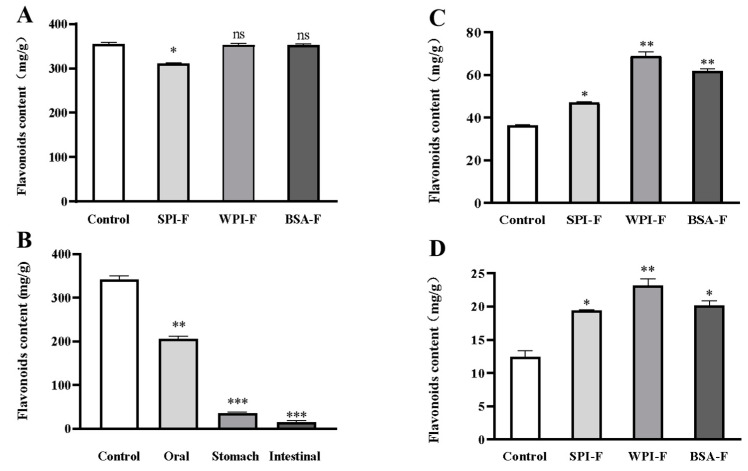
Changes of the flavonoids content after combination with different proteins. (**A**) the flavonoids content of the combined with different proteins; (**B**) the retained flavonoids content after the just the flavonoids being digested by using the simulated oral and gastrointestinal digestive solution; (**C**) the retained flavonoids content in the complex of proteins and flavonoids after being digested with the simulated gastric juice; (**D**) the retained flavonoids content in the complex of proteins and flavonoids after being digested with the simulated intestinal digestion. * *p* < 0.05, ** *p* < 0.01 and *** *p* < 0.001.

**Figure 3 ijms-24-06582-f003:**
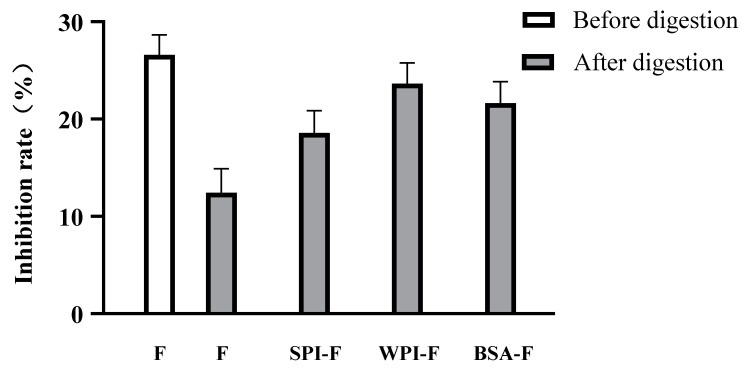
In vitro α-glucosidase inhibition rates of flavonoids with and without combied with protein before or after being digested with the simulated gastrointestinal conditions.

**Figure 4 ijms-24-06582-f004:**
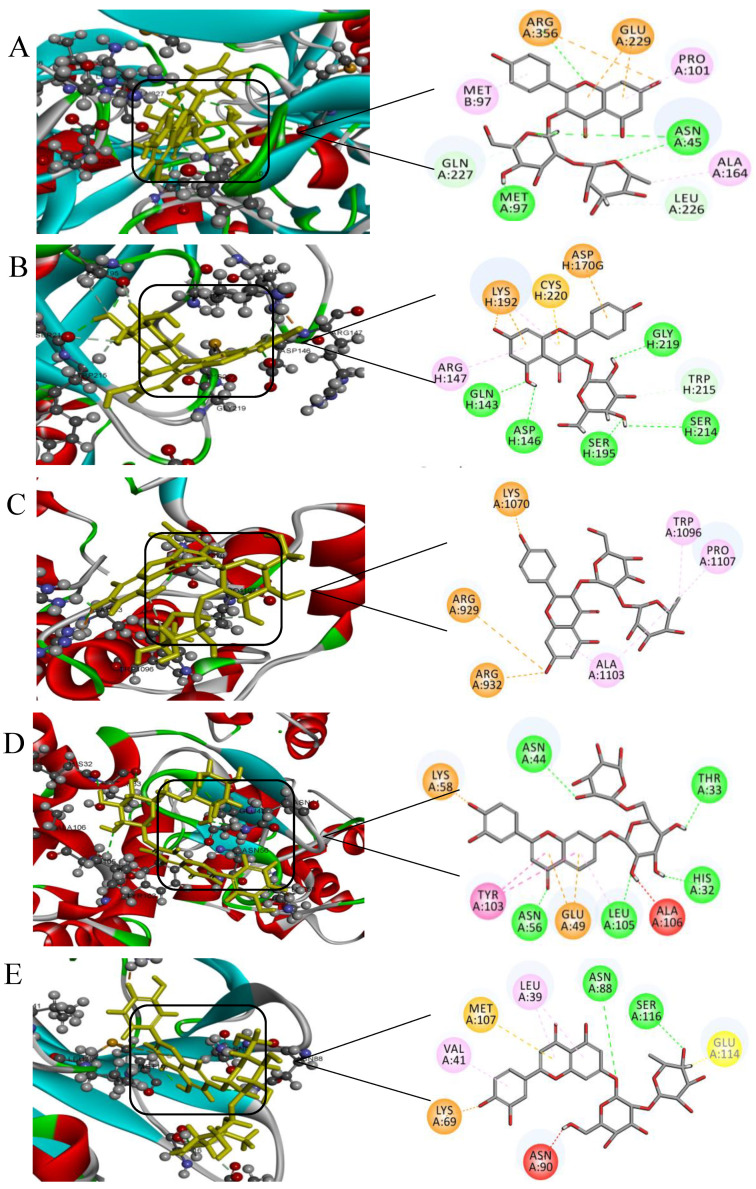
The complexe docking diagrams of the flavonoids with the different proteins. (**A**) the flavonoid (6)/SPI (7S), (**B**) the flavonoid (2)/SPI (7S), (**C**) the flavonoid (4)/BSA (**D**) the flavonoid(3)/α-La, (**E**) th flavonoids(5)/β-Lg.

**Figure 5 ijms-24-06582-f005:**
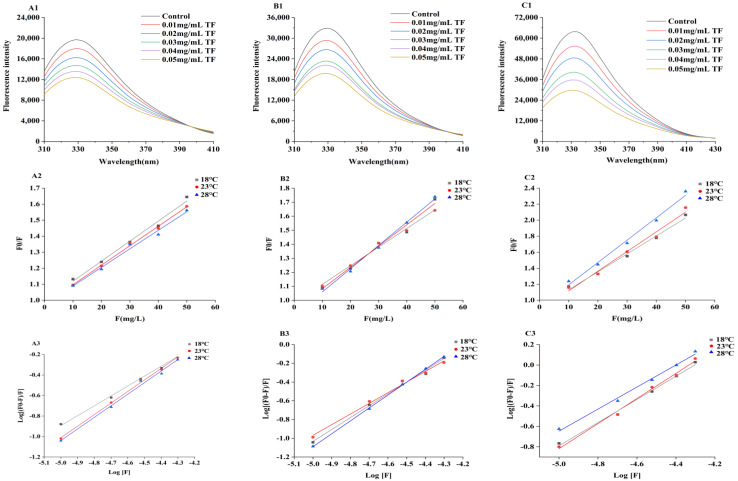
Fluorescence analysis of the interaction of proteins with flavonoids. (**A1**–**C1**): The fluorescence spectra of proteins with flavonoids; (**A2**–**C2**): The plots of log[(F0 − F)/F] versus log F at different temperatures; (**A3**–**C3**): The plots of F0/F versus F at different temperatures; (**A1**–**A3**): the combined protein was SPI-F; (**B1**–**B3**): the combined protein was WPI-F, (**C1**–**C3**): the combined protein was BSA-F.4.

**Table 1 ijms-24-06582-t001:** Component analysis of flavonoids in the ethanol extract of *A. keiskei*. The relative quantification in Table 1 represents the response intensity of characteristic ions in the mass spectrum, which is the value calculated by the area of the mass spectrum peak.

Compounds	Class	Relative Quantitative (%) Mean ± SD
Aureusidin-4-O-glucoside	Aurones	6.88 ± 0.11
Kaempferol-3-O-glucoside	Flavonols	4.62 ± 0.16
Luteolin-7-O-rutinoside	Flavones	4.22 ± 0.26
Kaempferol-3-O-neohesperidoside	Flavonols	4.04 ± 0.10
Luteolin-7-O-neohesperidoside	Flavones	4.04 ± 0.01
Kaempferol-3-O-glucorhamnoside	Flavonols	3.98 ± 0.08
Quercetin-5-O-β-D-glucoside	Flavonols	3.58 ± 0.11
6-C-MethylKaempferol-3-glucoside	Flavones	3.51 ± 0.02
Diosmetin-7-O-glucoside	Flavones	3.51 ± 0.04
Diosmetin-7-O-galactoside	Flavones	3.68 ± 0.05
Hispidulin-7-O-Glucoside	Flavones	3.42 ± 0.02
Diosmetin-7-O-rutinoside (Diosmin)	Flavones	2.10 ± 0.08
Luteolin-4′-O-glucoside	Flavones	2.36 ± 0.23
Quercetin-3-O-galactoside (Hyperin)	Flavonols	2.28 ± 0.08
Luteolin-7,3′-di-O-glucoside	Flavones	1.78 ± 0.07
Quercetin-3-O-glucoside (Isoquercitrin)	Flavonols	2.15 ± 0.05
Luteolin-3′-O-glucoside	Flavones	1.92 ± 0.02
Yuanhuanin	Flavones	1.55 ± 0.18
Isobavachalcone	Chalcones	1.73 ± 0.03
Xanthoangelol F	Chalcones	1.63 ± 0.05
Quercetin-7-O-glucoside	Flavonols	1.49 ± 0.02
Quercetin-4′-O-glucoside (Spiraeoside)	Flavonols	1.49 ± 0.03

**Table 2 ijms-24-06582-t002:** Molecular docking energy of six flavonoid monomers and five crystals. The five crystals are BSA (PDB ID: 3VO3), α-La (PDB ID:1HFZ), β-Lg (PDB ID: 3NPO), 11S protein (PDB ID: 1OD5) soybean and 7S protein (PDB ID: 3AUP), respectively.

Compounds	Docking Energy (kJ mol^–1^)					Docking Interaction Energy (kJ mol^−1^)				
	3VO3	1HFZ	3NPO	1O5D	3AUP	3VO3	1HFZ	3NPO	1O5D	3AUP
Aureusidin-4-O-glucoside	−11.6258	−7.5076	−1.8926	−16.3452	−15.0624	−50.1247	−52.6416	−46.8467	−68.3019	−53.0224
Kaempferol-3-O-glucoside	−14.2774	−26.1547	−29.1191	−22.3671	−13.7369	−53.6227	−50.7946	−57.8633	−65.2053	−55.6249
Luteolin-7-O-rutinoside	−10.6571	−28.2081	−25.8475	−23.1766	−12.6838	−50.6169	−59.0243	−51.0303	−79.6111	−70.0672
Kaempferol-3-O-glucorhamnoside	−4.8362	−16.1509	−20.2018	−23.4815	−6.8266	−57.3834	−63.6072	−62.1672	−75.4202	−57.5002
Kaempferol-3-O-neohesperidoside	−6.8933	−14.0788	−15.8156	−20.6677	−8.4797	−58.3429	−57.0833	−65.0644	−70.7281	−63.8887
Luteolin-7-O-neohesperidoside	−0.4392	−15.3671	−16.6973	−19.2924	−0.2389	−41.7946	−51.7348	−64.2016	−70.7572	−57.5002

**Table 3 ijms-24-06582-t003:** The molecular quenching constants (kq), thermodynamic binding constants (Ka), numbers of binding sites (n), and thermodynamic parameters for SPI-F, WPI-F, and BSA-F complexes at 18, 23, and 28 °C.

Protein	T	Kq	Ka	n	ΔH	ΔS	ΔG
	(°C)	(10^12^ L·mol^−1^s^−1^)	(10^5^L·mol^−1^)		(kJ·mol^−1^)	(J·mol^−1^·K^−1^)	(kJ·mol^−1^)
	18	1.23 ± 0.66	0.1486 ± 0.01	0.95 ± 0.14		104.38 ± 4.39	−23.23 ± 0.14
SPI	23	1.10 ± 0.94	0.1866 ± 0.01	1.15 ± 0.22	7.13 ± 0.71	110.26 ± 2.18	−25.50 ± 0.09
	28	1.03 ± 0.10	0.2245 ± 0.02	1.16 ± 0.13		110.40 ± 2.75	−26.09 ± 0.15
	18	1.21 ± 0.21	1.6867 ± 0.11	1.35 ± 0.05		178.16 ± 2.79	−28.25 ± 0.17
WPI	23	1.23 ± 0.12	2.8689 ± 0.02	1.36 ± 0.18	26.59 ± 0.63	188.19 ± 3.75	−29.11 ± 0.17
	28	1.19 ± 0.27	5.9921 ± 0.10	1.42 ± 0.29		198.94 ± 2.14	−33.29 ± 0.04
	18	1.85 ± 0.19	1.0525 ± 0.10	0.98 ± 0.18		133.91 ± 0.92	−28.73 ± 0.16
BSA	23	2.90 ± 0.30	1.4437 ± 0.22	1.13 ± 0.13	10.23 ± 0.37	132.31 ± 2.29	−28.93 ± 0.17
	28	1.43 ± 0.18	3.3640 ± 0.07	1.21 ± 0.18		138.02 ± 1.55	−31.31 ± 0.17

## Data Availability

The data presented in this study are available within the article, Figures, Tables, and Appendix A.

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
