# Peer review of "A Comparative Study of Binding Interactions between Proteins and Flavonoids in Angelica Keiskei: Stability, α-Glucosidase Inhibition and Interaction Mechanisms"

_ijms, 2023, doi:10.3390/ijms24076582_

Round 1

Reviewer 1 Report

General Comments:

This paper entitled “A Comparative Study of Binding Interactions between Proteins and Flavonoids in Angelica Keiskei: Stability, α-Glucosidase Inhibition and Interaction Mechanisms” reporting the flavonoids in Angelica Keiskei. The author demonstrated using soy protein isolate (SPI), whey protein isolate (WPI) and bovine serum albumin (BSA) as delivery systems, the stability and bioactivity of flavonoids in Angelica Keiskei were studied, they found the protection effect of total flavonoids of WPI-F provided a certain reference for the stability and biological activity research of total flavonoids in Angelica Keiskei. However, there are several grammatical and spelling errors as well as instances of badly worded/constructed sentences. Please check the manuscript and refine the language carefully. In conclusion, I endorse acceptance provided the authors would address the following minor remarks.

Question 1:

Line 31-33, "Our results show that the flavonoid content of WPI-F (23.17 ± 0.86 mg/g) was higher than those of SPI-F (19.41 ± 0.56 mg/g) and BSA-F (20.15 ± 0.62 mg/g) after simulated digestion in vitro", which is somewhat subjective.

Question 2:

What does P-F mean in the abstract?

Question 3:

Line 58, "a delivery system" was not a transmission path. That was a protein-based nanocomplexes. Please change another descriptive word.

Question 4:

Why was there no comparison of pre-digested SPI-F, WPI-F, and BSA-F for α-glucosidase inhibition rate?

Question 5:

Was SPI-F, WPI-F, and BSA-F inhibition of α-glucosidase conducted after simulated digestion in vitro? Does the process of digestion cause the protein to break down, thus affecting the inhibitory effect?

Question 6:

What is the main innovation of the article?

Question 7:

Why are SPI, WPI, and BSA selected as delivery carriers?

Question 8:

The flavonoids in A. keiskei are mixtures. How can we clarify the binding mechanism of flavonoids and proteins?

Question 9:

In this paper, the biological activities of flavonoids in A. keiskei were less studied.

Question 10:

The conclusion is the summary of this paper, it is better not to cite references for explanation.

Author Response

  1. Review comments and replies:

Question 1

Line 31-33, "Our results show that the flavonoid content of WPI-F (23.17 ± 0.86 mg/g) was higher than those of SPI-F (19.41 ± 0.56 mg/g) and BSA-F (20.15 ± 0.62 mg/g) after simulated digestion in vitro", which is somewhat subjective.

Response 1

The expression "Our result" is somewhat subjective. We have replaced "Our result" with "The result". (After revision line 20)

Question 2

What does P-F mean in the abstract?

Response 2

The full name of P-F in the abstract is protein-flavonoids. It is indicated in the article. (After revision line 25 and line 80)

Question 3

Line 58, "a delivery system" was not a transmission path. That was a protein-based nanocomplexes. Please change another descriptive word.

Response 3

"a delivery system" is indeed an inexact expression. "a delivery system" has been replaced with "protein-based nanocomplexes". (After revision line 45)

Question 4

Why was there no comparison of pre-digested SPI-F, WPI-F, and BSA-F for α-glucosidase inhibition rate?

Response 4

Figure 3. shows that the inhibitory rate of flavonoids (F) on α-glucosidase before and after digestion has been compared, and it is concluded that the inhibitory rate of flavonoids (F) on α-glucosidase after digestion has been reduced. In order to better conform to the topic of stability of SPI-F, WPI-F, and BSA-F in this paper, it is only necessary to compare the inhibitory activity of flavonoids (F) on α-glucosidase after binding to the three proteins.

Question 5

Was SPI-F, WPI-F, and BSA-F inhibition of α-glucosidase conducted after simulated digestion in vitro? Does the process of digestion cause the protein to break down, thus affecting the inhibitory effect?

Response 5

The inhibition of SPI-F, WPI-F and BSA-F on α-glucosidase was conducted in vitro after simulated digestion. The purpose of this experiment was to compare the inhibition rates of SPI-F, WPI-F and BSA-F on α-glucosidase after simulated digestion. Other conditions being equal, even if the protein breaks down, it doesn't affect the purpose of the experiment.

Question 6

What is the main innovation of the article?

Response 6

The main innovation of this paper is to compound WPI, SPI and BSA with flavonoids in A. keiskei to compare their binding ability, stability, and biological activity. So as to screen out the best protein-based nanocomplex. Further improved functional application of flavonoids in A. keiskei. 

Question 7

Why are SPI, WPI, and BSA selected as delivery carriers?

Response 7

WPI, SPI, and BSA come from milk, plant and animal respectively, which are representative to a certain extent. WPI, SPI, and BSA have good emulsification, solubility, easy absorption and high stability. These three proteins are widely used by many researchers as transport and encapsulation materials for various polyphenol compounds. In addition, these three proteins are also the most reported as delivery carriers.

Question 8

The flavonoids in A. keiskei are mixtures. How can we clarify the binding mechsanism of flavonoids and proteins?

Response 8

The flavonoid compounds in A. keiskei are mixtures. It is difficult to elucidate the interaction mechanism between proteins and mixtures, which is the limitation of this paper. However, a large number of papers have reported the mechanism of interaction between proteins and different flavonoid monomers. Therefore, in this paper, six flavonoid monomers with relatively high content in the mixture were used for molecular docking with proteins to further help understand the mechanism of interaction.

Question 9

In this paper, the biological activities of flavonoids in A. keiskei were less studied.

Response 9

A previous article was published to study the hypoglycemic and lipid-lowering activity of flavonoids in A. keiskei. In this paper, we compare the stability of three protein-based nanocomplexes. On this basis, the main purpose of this paper is to compare the physiological activities of protein-based nanocomplexes by the inhibition of α-glucosidase and screen out the best nanocomplex.

Question 10

The conclusion is the summary of this paper, it is better not to cite references for explanation.

Response 10

References and corresponding text analysis have been deleted from the conclusion.

Reviewer 2 Report

Dear Authors,

Thank you very much for the opportunity to review your manuscript entitled “A Comparative Study of Binding Interactions between Proteins and Flavonoids in Angelica Keiskei: Stability, α-Glucosidase Inhibition and Interaction Mechanisms”. The article describes the evaluation of whey protein isolate (WPI), soybean protein isolate (SPI), and bovine serum albumin (BSA) on flavonoid stability and activity – particularly after being exposed to in vitro gastric digestion conditions – and the in silico predictions of molecular docking studies. The manuscript is very well written and explained, and the work significantly expands upon the literature regarding the bioavailability and transport of biomedical compounds. One major point and a few minor suggestions are listed below. Any consideration in these items would be greatly appreciated. Congratulations on the very nice work!

Minor:

1) I’m not sure if this is just a personal preference, but I’d include the full name and acronyms of “whey protein isolate (WPI), soybean protein isolate (SPI), and bovine serum albumin (BSA)” in the Introduction in addition to the abstract.

2) Please include the legend into the x-axis on Figures 2C and 2D

3) A bar graph may be more appropriate for Figure 3, since the x-axis groups are not connected to one another.

4) What are the differences in predicted affinity between the proteins and docked ligands? I believe AutoDock has a reported score, but using a web server like PRODIGY (https://wenmr.science.uu.nl/prodigy/) might also be helpful.

Major:

1) How much do the flavonoids contribute to the decreased fluorescence independent of their effects on protein binding? Have the authors controlled for this (the changes in measured fluorescence after the addition of flavonoids) by measuring the flavonoids in solution with no protein?

Round 2

Reviewer 1 Report

Accept